# Feasibility randomised multicentre, double-blind, double-dummy controlled trial of anakinra, an interleukin-1 receptor antagonist versus intramuscular methylprednisolone for acute gout attacks in patients with chronic kidney disease (ASGARD): protocol study

Gowrie Balasubramaniam,[1] Trisha Parker,[2] David Turner,[3] Mike Parker,[2] Jonathan Scales,[4] Patrick Harnett,[1] Michael Harrison,[2] Khalid Ahmed,[5] Sweta Bhagat,[6] Thiraupathy Marianayagam,[7] Costantino Pitzalis,[8] Christian Mallen,[9] Edward Roddy,[9] Mike Almond,[1] Bhaskar Dasgupta[1]

► Prepublication history and additional material are available. To view these files, please visit the journal online (http://dx.doi.org/10.1136/bmjopen-2017-017121).

For numbered affiliations see end of article.

**Correspondence to**
Dr Gowrie Balasubramaniam;
gowrie@doctors.net.uk

## ABSTRACT

**Introduction** Acute gout occurs in people with chronic kidney disease, who are commonly older people with comorbidities such as hypertension, heart disease and diabetes. Potentially harmful treatments are administered to these vulnerable patients due to a lack of clear evidence. Newly available treatment that targets a key inflammatory pathway in acute gout attacks provides an opportunity to undertake the first-ever trial specifically looking treating people with kidney disease. This paper describes the protocol for a feasibility randomised controlled trial (RCT) comparing anakinra, a novel interleukin-1 antagonist versus steroids in people with chronic kidney disease (ASGARD).

**Methods and analysis** ASGARD is a two-parallel group double-blind, double-dummy multicentre RCT comparing anakinra 100 mg, an interleukin-1 antagonist, subcutaneous for 5 days against intramuscular methylprednisolone 120 mg. The primary objective is to assess the feasibility of the trial design and procedures for a definitive RCT. The specific aims are: (1) test recruitment and retention rates and willingness to be randomised; (2) test eligibility criteria; (3) collect and analyse outcome data to inform sample and power calculations for a trial of efficacy; (4) collect economic data to inform a future economic evaluation estimating costs of treatment and (5) assess capacity of the project to scale up to a national multicentre trial. We will also gather qualitative insights from participants. It aims to recruit 32 patients with a 1:1 randomisation. Information from this feasibility study will help design a definitive trial and provide general information in designing acute gout studies.

**Ethics and dissemination** The London-Central Ethics Committee approved the protocol. The results will be disseminated in peer-reviewed journals and at scientific conferences.

### Strengths and limitations of this study

► Assessing the feasibility of undertaking a definitive robustly designed double-blind, double-dummy study.
► Qualitative aspects to current study design will help future study be more patient orientated.
► Current study is not designed to find differences in outcomes.

**Trial registration number** EudraCT No. 2015-001787-19, NCT/Clinicalstrials.gov No. NCT02578394, pre-results, WHO Universal Trials Reference No. U1111-1175-1977. NIHR Grant PB-PG-0614–34090.

## INTRODUCTION
### Background and rationale

Chronic kidney disease (CKD) affects 5% of the UK population[1] and 40% of patients with CKD 3 and 4 have chronic gout,[2] suggesting that 1.32 million people, predominantly older patients, have CKD and gout in the UK. The overall incidence of acute gout attacks has been estimated to be approximately 2 per 1000 person-years.[3,4] The prevalence of gout is increasing due to increasing prevalence of comorbid conditions that are associated with hyperuricaemia such as hypertension, obesity, metabolic syndrome, diabetes and chronic kidney disease.[4,5]

In a UK study, approximately 89.4% of patients were treated for their acute attack with non-steroidal anti-inflammatory agents

(NSAIDS) which are well known to worsen kidney failure.[5] Secondary care data from the USA showed that up to 50% of acute gout patients had CKD and 40% were given contraindicated medications.[6] This is similar to our experience where a review of inpatient acute gout care showed half of all patients with acute gout and kidney disease were given potentially harmful medications. Mismanagement of acute gout attacks in patient with kidney disease occurs as there is no firm evidence base for treating acute gout attacks in people with kidney disease.[7]

The lack of evidence in the literature for the use of conventional agents in patients with CKD is primarily because patients with CKD are excluded as NSAIDS are used as an active comparator in randomised controlled trials (RCTs). Guidelines continue to suggest using colchicine with in patients with renal disease even with its toxicity profile and poor evidence base.[8–10] We now have an opportunity to conduct a trial using two agents that would not be expected to have adverse effects on renal function.

### Treatment options for patient with acute gout attacks and CKD

NSAIDS have the largest evidence base for use in acute gout but are contraindicated in CKD as they can cause an acute kidney injury. They can also worsen heart failure, hypertension, liver failure and cause gastrointestinal bleeding.[7 11] Colchicine is an alternative anti-inflammatory agent, it has a narrow therapeutic index and a low-dose regime is advocated following findings in the high versus low dosing of oral colchicine for early acute gout flare (AGREE) trial. Even though this study excluded patients with moderate to severe kidney disease (glomerular filtration rate (GFR) <60 mL/m$^3$), 23% of patients on the low-dose treatment still experienced diarrhoea.[12] Diarrhoea is more common in patients with CKD, they are also more likely to have drug accumulation and serious side effects from toxicity such as bone marrow failure, rhabdomyolysis, pancreatitis and myopathy.[7]

Glucocorticoids are an effective treatment with some evidence for use in patients with CKD from older trials. Triamcinolone, a long-acting synthetic corticosteroid, was shown to be as effective as adrenocorticotrophic hormone, a hormone that increases endogenous cortisol secretion.[11] A more recent double-blind randomised controlled study[13] showed that prednisolone 35 mg for 5 days was not inferior to naproxen for acute gout attacks, although patients with moderate to severe renal impairment were excluded.

### Interleukin-1 inhibitors

Interleukin-1 (IL-1) signalling has been shown to play a key role in gout-associated inflammation[14 15] and targeted therapy is now available. Two agents of IL-1 have been used in RCTs. Rilonocept, an IL-1 decoy receptor, has been shown to be efficacious in reducing acute flares during the initiation of urate lowering therapy,[16 17] but no added benefit was obtained for the treatment of acute gout when it was used in addition to an NSAID.[18]

Canakinumab, a monoclonal antibody against IL-1 beta, showed efficacy preventing acute flares during the initiation of urate-lowering therapy[19] and for acute gout in patients with difficult disease or contraindication to colchicine or NSAIDs.[20–22] It was associated with increased adverse events after a single injection[23] which was felt in part due to its prolonged action which can be for many weeks.[24] This study will use anakinra (Kineret), an IL-1 receptor antagonist that competes with IL-1beta for its receptor. A pilot study of its use in gout showed prompt resolution of symptoms in patients treated with anakinra,[25] including in three patients with renal impairment. There have been further reports of the use of anakinra in difficult to treat cases showing good efficacy, with most patients having a good response within 24 hours.[26–31]

Anakinra has a good pharmacokinetic profile with a shorter duration of action, accumulation can occur in severe renal failure,[32] and this had been linked to an increased risk of infection in one case report.[31] The dose of a daily 100 mg has been used in studies looking at treatment of rheumatoid arthritis (usually between 50 and 150 mg), and it was well tolerated for up to 3 years.[33–37] The licensed use of anakinra is for Cryopryin Associated Periodic Syndrome where dosing is 3–4 mg/kg in severe disease (doses up to 800 mg).[38] The rationale for the duration of treatment use of 100 mg subcutaneous for 5 days come from emerging data where anakinra has been used to treat acute gout with good efficacy and in patients with CKD without too many adverse events.[25–29] Some reports vary between 3 to 5 days but most use 5 days' duration of treatment to ensure adequate response in all patients, including our own experience.[30] We have intentionally avoided using anakinra in patients with severe renal failure (GFR <30 mL/min/1.73 m$^2$) although it has been used safely in patient on haemodialysis using an alternate regime,[39] and we have some experience of this in our centre.[30] Anakinra is much cheaper compared with canakinumab (£131.15 versus £9927.90 for a course of treatment), which has been given a license in the UK for use in difficult to treat gout.

Safety experience stem mainly from studies in rheumatoid arthritis where patients were on concurrent immunosuppressive treatment and the duration of treatment was prolonged (up to 6 months). A slightly higher risk of infection (2%) and neutropaenia was noted. We do not anticipate these issues as the treatment time for acute gout will be short, 5 days versus many months, and we feel the risk of infection should be much less. We are excluding people with concurrent immunosuppressive treatment and will exclude people with serious infection. There are some data of the use of anakinra in the setting of an active infection from the first clinical trial where it was used to treat people with severe septicaemia and those who received anakinra were not worse off.[40] There are also reports of the use of anakinra for acute gout for hospitalised inpatients with active infections and multiple comorbidities.[27–29] There was one case report of neutropaenia in one patient with a kidney transplant with severe

renal impairment.[31] Blood counts will be checked at the end of treatment at day 5 and participants will also be assessed for infection face to face on day 2, day 7 and 8 weeks, and by telephone contact on day 3, day 4 and day 5. Participants will be asked to stop treatment if serious infection develops.

## Methylprednisolone

Methylprednisolone acetate is one of the most commonly used long-acting intramuscular agent in the UK, we have shown this in our region,[41] and it is equivalent to Triamcinolone in other joint conditions.[42]

We are using methylprednisolone 120 mg given by intramuscular injection which is as efficacious as oral steroids with a lower cumulative dose (120 mg vs 175 mg) in other rheumatology conditions, and this is equivalent to the dose of oral prednisolone used in the trial by Janssens et al[13] (35 mg for 5 days). This well-designed double-blind randomised controlled study showed that prednisolone 35 mg for 5 days was not inferior to naproxen. The rate

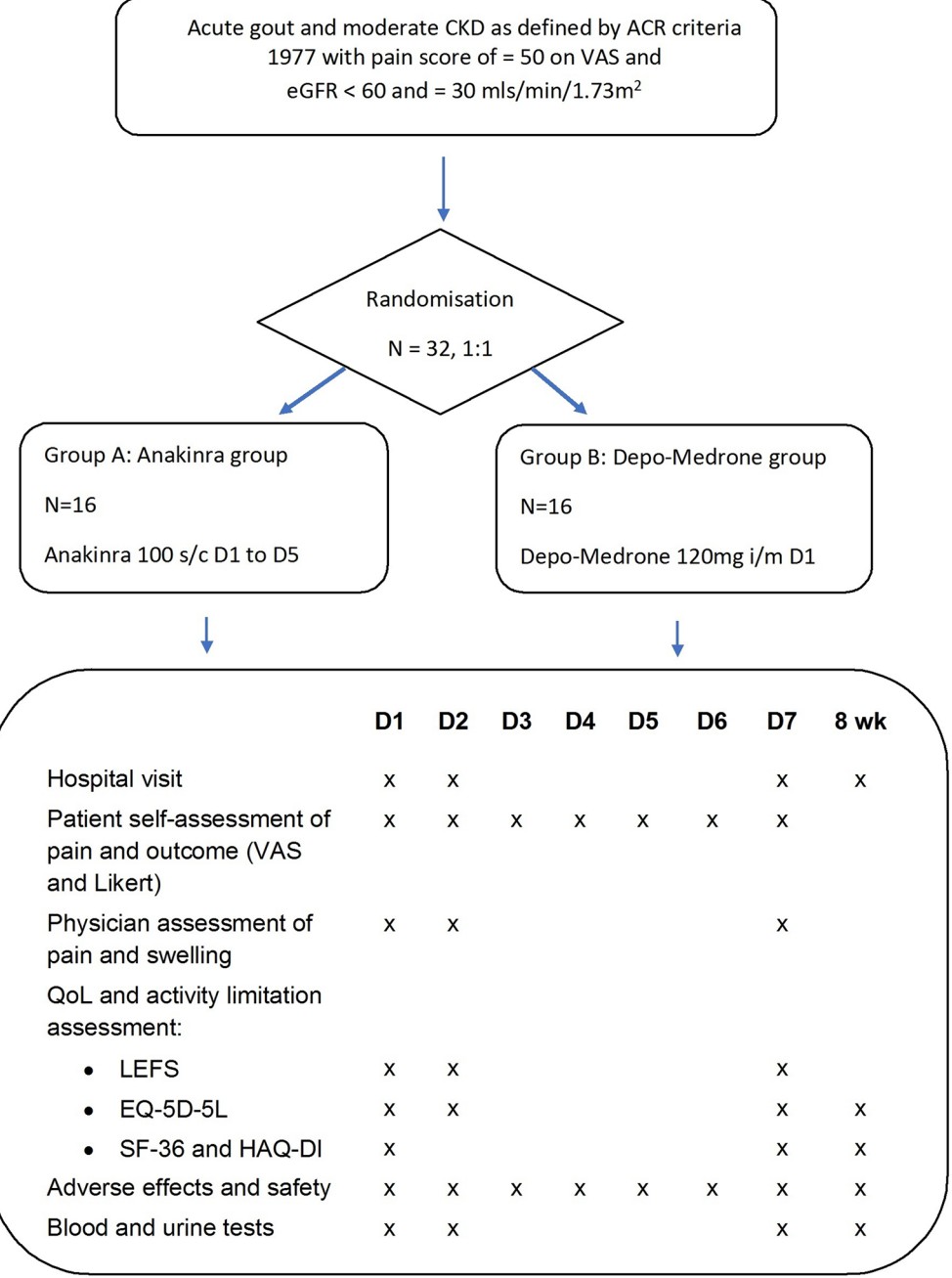

**Figure 1**  ASGARD study flowchart. Randomised participants will receive allocated treatment and placebo equivalent. ACR, albumin creatinine ratio; ASGARD, interleukin-1 antagonist versus steroids in people with chronic kidney disease; CKD, chronic kidney disease; EQ-5D-5L, Five-level EuroQol five-dimensional questionnaire HAQ-DI, Health assessment questionnaire disability index; LEFS, Lower Extremity Functional Scale; QoL, quality of life; SF-36, 36-Item Short Form Survey; VAS, Visual Analogue Scale.

---

**Box 1   Anakinra versus steroids for gout attacks in patients with chronic renal disease inclusion criteria**

Patients who meet inclusion criteria will be approached by a study investigator. Consent for participation will be sought. On gaining consent, the investigator will go through exclusion criteria.
1. Subjects capable of giving informed consent;
2. Male or non-pregnant, non-nursing female;
3. ≥18 years of age;
4. Estimated glomerular filtration rate <60 mL/min/1.73 m² and ≥30 mL/min/1.73 m² as calculated using serum creatinine and modified Modification of Diet in Renal Disease Study formula or Chronic Kidney Disease Epidemiology Collaboration on two occasions at least 2 months apart with one being in the last 6 months. Creatinine at time of presentation can be used unless participant has an acute kidney injury as defined by serum creatinine rise by ≥26 µmol/L within 48 hours or ≥1.5-fold rise from baseline value.
5. Diagnosis of acute gout arthritis as defined by the American College of Rheumatology 1977 preliminary criteria;
6. Gout attack less ≤36 hours;
7. Baseline pain intensity greater than or equal to 50 mm on the 0–100 mm Visual Analogue Scale. In the case of multiple joints (≤3), the most affected joint will be assessed.

---

of side effects even with short-term steroid treatment in acute gout is unknown. Our study may help establish steroids, especially intramuscular methylprednisolone which has limited evidence base, as a potential safe, cheap and readily available treatment for acute gout in patients with renal disease. If the initial results show equivalent efficacy between the two treatment arms, then we could go on to establish steroids as the standard treatment and make considerable savings.

Our use is only as a one-off injection (as is commonly done in routine practice), and we do not anticipate long-term complications. Short term side-effects we may encounter are uncontrolled blood sugars in people with diabetes and worsening oedema in people with heart failure and kidney failure. The latter should occur less with methylprednisolone which is associated with less mineralocorticoid effect than other steroids (eg, prednisolone). The placebo equivalent for the methylprednisolone will be Lipofundin, a lipid emulsion commonly used in parenteral nutrition but has been use as a placebo agent in other trial using steroids.

## Objectives
### Primary objective
The primary objective of interleukin-1 antagonist versus steroids in people with chronic kidney disease (ASGARD) is to assess the feasibility of the trial design and procedures for a definitive RCT. The specific aims are: (1) test recruitment and retention rates and willingness to be randomised; (2) test eligibility criteria; (3) collect and analyse outcome data to inform sample and power calculations for a trial of efficacy; (4) collect economic data to inform a future economic evaluation estimating costs of treatment and (5) assess capacity of the project to scale up to a national multicentre trial. We will also gather qualitative insights from participants about their experience from being in the feasibility trial to help design the subsequent definitive trial.

### Secondary objectives
Information will be obtained to help with the design of the subsequent study including testing of proposed primary and secondary outcomes described below.

### Trial design
ASGARD is a two-parallel group double-blind, double dummy multicentre RCT with a 1:1 allocation ratio comparing anakinra 100 mg subcutaneous for 5 days with a one-off methylprednisolone (Depo-Medrone) 120 mg intramuscular injection (figure 1) (see online supplementary material).

## METHODS
### Study setting
ASGARD aims to recruit from centres in the East of England region. Recruitment will be from patients who present to primary and secondary care with acute gout and chronic kidney disease, this will also include inpatients who develop an acute gout attack. Known patients with moderate kidney disease will also be primed to contact the research team by working with local primary care providers to send out letters of invitation. Recruitment will occur in the secondary care setting with treatment occurring on an outpatient basis or can occur in secondary care if symptoms warrant admission or if the participant is already an inpatient and develops an acute gout attack.

### Eligibility criteria

### Baseline assessments
Baseline assessment will include basic medical observations, routine bloods and basic clinical examination (boxes 1 and 2). The 2015 American College of Rheumatology/European League Against Rheumatism Collaborative Initiative will also be used at baseline assessment.[43]

### Interventions
Eligible participants will undergo 1:1 block randomisation to one of two treatment arms; anakinra 100 mg s/c daily for 5 days and one-off placebo methylprednisolone (Lipofundin MCT) intramuscular (gluteal) or one-off intramuscular methylprednisolone (Depo-Medrone) and placebo anakinra subcutaneous for 5 days. The study drug and its placebo will be provided by SOBI. Lipofundin will be purchased by the coordinating centre pharmacy by the usual route.

For anakinra and its equivalent placebo, participants will be issued a pharmacy box with individual syringes in syringe holders and a sharps bin. Each syringe will be labelled day 1 (already administered) up to day 5. Day 1 treatment will be administered after baseline assessment and investigations.

1.  Treatment with colchicine in the last week;
2.  Initiation of or change to dose of non-steroidal anti-inflammatory drugs (NSAIDs) in the last week. NSAID use is not an exclusion criteria if taken at a stable dose for at least 4 weeks;
3.  Initiation of or change to dose of systemic steroids in the last week. Systemic steroid use is no an exclusion criteria if taken at an equivalent dose of 10 mg prednisolone or less for the last 4 weeks;
4.  Polyarticular gout, that is, affecting four or more four joints;
5.  Rheumatoid arthritis, evidence/suspicion of infectious/septic arthritis or other acute inflammatory arthritis such as systemic lupus erythematosus, mixed connective tissue disease, scleroderma, polymyositis or significant systemic involvement secondary to rheumatoid arthritis.
6.  Concurrent immunosuppression/immunomodulatory treatment (calcineurin inhibitor, antiproliferative or biological) therapy for other reason, that is, organ transplant;
7.  History or current inflammatory joint disease other than gout (eg, rheumatoid arthritis, reactive arthritis, psoriatic arthritis, seronegative spondyloarthropathy, Lyme disease);
8.  Current active malignancy (with the exception of basal cell or squamous cell carcinoma of the skin, cervical intraepithelial neoplasia and non-metastatic/advanced prostate cancer);
9.  Any patients with contraindication to intramuscular injection such as coagulopathy or thrombocytopaenia (platelet count <100×109/L (100,000/mm$^3$));
10. Abnormal liver function tests: total bilirubin >upper limit of normal, alanine aminotransferase or aspartate aminotransferase >2 times upper limit of normal;
11. Haemoglobin<85 g/L (8.5 g/dL);
12. White blood cell count <1.5×109/L (1000/mm$^3$), absolute neutrophil count <1.5×10$^9$/L (1000/mm$^3$);
13. Evidence of serious active infection requiring intravenous antibiotic drug administration within last 4 weeks. Potential participants who were empirically treated with a course of intravenous antibiotics for a presumptive diagnosis of an infection may be eligible where the treating medical team has subsequently excluded an active infection and consequently discontinued antibiotic therapy at least 4 hours before randomisation. These potential participants can be discussed with the investigator where clarity is required;
14. Evidence of uncontrolled concomitant cardiovascular, nervous system, pulmonary (including obstructive pulmonary disease), hepatic, endocrine (including uncontrolled diabetes) or gastrointestinal disease. Potential participants who have active concomitant disease can only be eligible after discussion and agreement with the treating medical team;
15. Known positive hepatitis B virus surface antigen, hepatitis C antibody or HIV.
16. Females of childbearing potential who are not willing to use highly effective birth control methods from the time of consent to 1 week after treatment discontinuation. Highly effective method of contraception (hormonal or barrier method of birth control, abstinence) consist of:
17. Combined (oestrogen and progestogen containing) hormonal contraception associated with inhibition of ovulation; oral, intravaginal, transdermal;
18. Progestogen-only hormonal contraception associated with inhibition of ovulation; oral, injectable, implantable;
19. Intrauterine device or intrauterine hormone-releasing system;
20. Bilateral tubal occlusion or vasectomised partner;
21. Sexual abstinence, defined as true abstinence when it is in line with the preferred and usual lifestyle of the participant. Periodic abstinence (eg, calendar, ovulation, sympthothermal, postovulation) and withdrawal methods are not acceptable methods of contraception;
22. Men who are sexually active with a female partner who are not willing to use highly effective birth control methods from the time of consent to 1 week after treatment discontinuation;
23. Females of childbearing potential must have a negative pregnancy test (highly sensitive urine or serum pregnancy test after a confirmed menstrual period) within 7 days prior to treatment initiation. Subjects are considered not of childbearing potential if they are surgically sterile (ie, they have undergone a hysterectomy, bilateral tubal ligation or bilateral oophorectomy) or they are postmenopausal.
24. Females must not be breastfeeding;
25. Patients who have had treatment as part of this trial cannot have repeat treatment for another flare as part of the trial;
26. Patients with allergies to excipients of IMPs: citric acid, anhydrous, sodium chloride, disodium edetate dehydrate, polysorbate 80, sodium hydroxide. Hypersensitivity to *Escherichia coli*-derived proteins, egg proteins and soy proteins. Patients with a latex allergy are also not eligible as the inner needle cover of the prefilled syringe contains dry natural rubber (a derivative of latex).

The participant will be taught to self-administer the subcutaneous injection. Participant will then attend on day 2 for assessment of self-injection and subsequently will undertake self-injection themselves on days 3, 4 and 5. Participants unable to self-inject will receive their injections from a carer, family member or district nurse.

Intramuscular methylprednisolone or Lipofundin syringes will be prepared by a member of the research team not involved with the trial assessments (ie, unblinded research nurse).

## Modifications

Injection sites will be assessed visually on day 2 and day 7 (figure 2), and by telephone contact from day 3 to day 6. Blood tests will be taken on day 2 and day 7 to check for neutropaenia and any signs of infection. Treatment will be stopped in the event of any adverse reaction during subcutaneous injections, oral prednisolone 35 mg orally for 5 days will be used as rescue treatment if symptoms of acute attack persist. Side effects to methylprednisolone such as worsening oedema and uncontrolled diabetes (if diabetic) will be monitored and reported as adverse events.

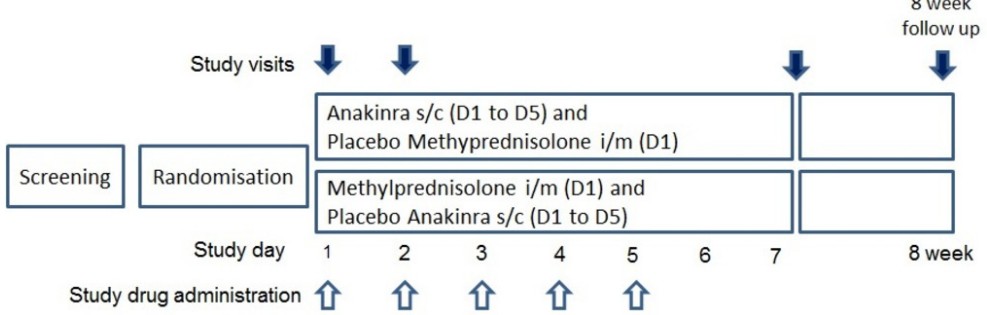

**Figure 2** Anakinra versus steroids for gout attacks in patients with chronic renal disease participant timeline. Participants will be assessed by research team at day 1 (randomisation), day 2, day 7 and 8 weeks.

### Adherence

Participants will be contacted everyday up to from day 3 to day 7, to remind them of injection up to day 5 and of diary entry up to day 7 (except over the weekend). Syringe holders will be accounted for and the number of syringes returned in the sharps box will be counted through the lid (not strictly) at day 7 before permanently locked by the trial team for disposal.

### Concomitant care

#### Rescue medications

Participants can take analgesia listed in the protocol as non-investigational medicinal product (co-dydramol, codeine phosphate, tramadol and paracetamol) for relief as required, usage will be recorded. If participants get another flare during treatment, this will be recorded and a coarse of oral prednisolone 35 mg orally daily for 5 days will be used for rescue treatment. Low-dose colchicine 500 μg orally twice daily could be used in addition. Use of other immune-modifying treatment or NSAIDS is prohibited.

### Outcomes

This is a feasibility study. It is not powered to look for inference. Study processes will be analysed. Recruitment and retention rates and willingness of patients to be randomised will be calculated. The proportion of patients who did not meet eligibility criteria will be examined. Adherence and compliance rates and qualitative feedback will be examined. Economic data on healthcare resource use and health-related quality of life will be collected. Safety outcome measures will be reported as mandated in a clinical trial of an investigational medicinal product protocol.

### Primary endpoint/outcome

Feasibility of undertaking a definitive multicentre, double-blind, double dummy RCT to obtain clear guidance on the safe management of acute gout attacks in patients with chronic kidney disease.

Information from proposed outcome measures for the larger study will be sought. Proposed primary outcome measures of effectiveness consist of resolution of pain that is, time to 50% reduction and complete resolution of pain in self-assessed pain intensity in the joint most affected at baseline measure on the Visual Analogue Scale (0–100 mm) and 5-point Likert scale from baseline to 7 days' postrandomisation (day 1, day 2, day 3, day 4, day 5, day 6, day 7) using composite time points kept to 24 hours intervals as close as possible.

Proposed secondary outcome measures will consist of patient reported outcome measure (day 1, day 2, day 3, day 4, day 5, day 6 and day 7), physician assessment of joint tenderness and swelling (day 1, day 2 and day 7), assessment of activity limitation and quality of life: Health Assessment Questionnaire Disability Index and 36-Item Short Form Survey at day 1, day 7 and 8 weeks; Five-level EuroQol Five-Dimensional Questionnaire at day 1, day 2, day 7 and 8 weeks; Lower Extremity Functional Scale at day 1, day 2, day 7. Time to take rescue medication and a limited assessment of healthcare resource will be made.

Blood and urine tests renal on day 1, day 2, day 7 and 8 weeks, consisting of full blood count, urea and electrolytes, estimated GFR glucose, C reactive protein, serum uric acid and spot urine uric acid.

### Participant timeline

#### Sample size

This is a feasibility study and we will aim to recruit at least 16 patients in each arm of the study. The sample size was obtained from a recommendation based on the feasibility of running a parallel group design. The study will aim to enrol for 15 months.

#### Recruitment

Recruitment will be from secondary care, either when potential participants present with acute gout attack or if in-patients develop acute gout attack. Trial centres will be encouraged to work with primary care centres to identify potential participants and give letter of invitation. We will use ethically approved advertising material in the community. Potential participants will be given or sent a participant information sheet (see online supplementary material).

Our hospital serves a population of 350 000 and had an average of 47 patients per year over the last 4 years with approximately 50% of patients having chronic kidney

disease (ie, 23 patients per year). Enrolment is for 15 months from six centres, we will only recruit patients in working hours, that is, 9:00–17:00, it would be logistically difficult to avoid delays to treatment if participants present out of hours and face-to-face assessments cannot be done at weekends due to lack of research infrastructure in most small to medium-sized hospitals. All hospitals should be able to have pharmacy support to provide preparation and recording of treatment administered.

Potential participants would hopefully be fully aware of study if a letter of invitation was already sent to them. Otherwise, potential participants can have up to 24 hours to decide if they wish to take part in the study. Analgesia that is listed can be used if required; however, baseline assessments can only occur 4 hours after administration of last dose, that is, before the administration of the next dose. Treatment cannot be withheld during this period and exclusion criteria will be applied if duration of attack is longer than accepted or if contraindicatory medication is administered.

## Allocation
### Sequence generation
Enrolled patients will undergo 1:1 randomisation in blocks of four. Randomisation will be undertaken using an internet-based randomisation system within Anglia Ruskin Clinical Trial Unit (ARCTU). The ARCTU uses the Trans European Network Alea (TENALEA) system provided by the Trans European Network for Clinical Trial Service.

### Concealment mechanism
Once a patient has consented to take part in the trial, the designated staff will log in to the TENALEA web page to confirm eligibility and a random allocation will be sent to pharmacy. All investigators involved with the trial will be blinded to treatment. Patients will not be informed of their assigned treatment during the study. Pharmacy will keep a record of allocated treatment arms in the event emergency unblinding is required.

### Implementation
Anakinra and its placebo will be supplied by the manufacturer. Intramuscular methylprednisolone and its placebo (Lipofundin) will be prepared by a research team member not involved with the study.

Study labels will be provided to preserve blinding. Anakinra and its equivalent placebo will be provided by the manufacturer, once allocation is known, pharmacy will use label stating 'Anakinra or placebo' using standard operating procedure. Five individual syringes labelled 1 to 5 will be placed individual transparent syringe holders and packed for dispensing with a local pharmacy label. For intramuscular methylprednisolone (Depo-Medrone) and Lipofundin, a member of the study team not involved with the trial will draw the treatment and label with a study label stating 'Depomedrone or placebo,' using a standard operating procedure.

## Blinding
All trial investigators will be blinded to the intervention. Pharmacy will keep a record of treatments administrated. All trial participants, care providers and outcome assessors will be blinded to the treatment. Each treatment has its equivalent placebo to ensure blinding in maintained throughout study.

## Emergency unblinding
The treatment code for a participant can be broken by any clinician either directly or via contact of the principal investigator, allocation lists are available to the site pharmacy will provide 24-hour cover. Failing that, the central pharmacy can access allocation list. Where possible the local investigator should aim to discuss the need for unblinding with the coordinating investigator and blind should be endeavoured to be preserved to relevant research staff (data collection, analysis and interpretation). The coordinating investigator is responsible for pharmacovigilance management and reporting.

## METHODS: DATA COLLECTION, MANAGEMENT, ANALYSIS
### Data collection methods
Data collection will be by electronic clinical research forms (eCRF), these can be printed out if required for convenience of data collection (and act as source document) and then data inputted onto eCRF. Data can also be inputted direct onto eCRF and the completed eCRF can be printed out to act as source documents.

### Training plans
Study investigators must be aware of the classification criteria for gout. For the purposes of this study, each principal investigator will be asked to go through a presentation that will be available online and asked to attend study days where concerns will be raised.

### Quality control of data and results
Entered data will continuously reviewed by the data manager and any gaps in data will be fed back to the centre trial team. Routine blood results will be taken from the National Health Service (NHS) system. Samples for additional testing will be transported as per standard operating protocol to the co-ordinating centre.

### Participant retention and withdrawal
Enrolled participants will be contacted daily (weekdays) by the study team for a week. Participants may withdraw from the study for any reason at any time. The investigator may also withdraw participants from the study to protect their safety and/or if they are unwilling or unable to comply with required study procedures after discussion with the chief investigator. Participants who ask to stop study treatment, become pregnant or develop a serious infection (necessitating intravenous antibiotic treatment and hospital admission) should continue to attend all follow-up study. Participants who withdraw their consent at any point will be asked of their wish regarding the use

of data collected up to the point of withdrawal. Participants who do not attend follow-up assessment after site staff have attempted to contact the patient at least twice, for example, by telephone may be considered for withdrawal, the data already collected for them may be used and therefore needs returning in the usual manner. The PI responsible for a patient may choose to withdraw a patient from a trial for appropriate medical reasons, be they individual adverse events or new information gained about a treatment.

## Data management
Information from the patient diary and functional assessments will be entered onto an eCRF. Data will be entered as it is possible to do so that is, day 1, day 2, day 7 and 2 months. Referential data rules, valid values and range checks will be support by the data entry software. The data manager will review the data being sent at regular intervals and report back to the centre if there is any discrepancy. All forms, hard drives and storage devices related to the study will be kept in locked cabinets. Access to the study data will be restricted; a password system will be used to control access. Data will be stored at MACRO database.

## Study status reports
Regular updates of the study will be sent to the principle investigators via email and will be posted on the study website.

## Statistical methods
The sample size will be too small for estimates to have adequate precision, and some methods will not be technically possible for some patterns of missing values. To make use of data obtained, comparison of groups with the use of time-to-event ('survival') analysis will be undertaken where relevant. Comparison of continuous outcome measures will be undertaken with repeated measures analysis of variance with permutation tests. Two-sample comparisons of means will be by the two-sample permutation test. The association between the intervention and categorical variables will be tested using Fisher's exact test.

## Economic analysis
### Economic evaluation
Formal economic evaluation looking at the incremental cost-effectiveness of anakinra compared with the comparator is not planned as this is a feasibility study. We will collect relevant data on resource use (costs) and health-related quality of life to inform subsequent large-scale study. Key drivers of costs, resource allocation and scalability will be determined.

## Qualitative analysis
Twelve participants who give their specific informed consent for participation in the qualitative study will selected for an interview around their last 8 week follow-up. The interview will cover areas such as how participants became involved with the trial, the information that was provided for them, the process of obtaining informed consent and any areas that felt was not adequately covered or could have been undertaken differently.

## Data monitoring
The Trial Steering Committee will consist of the chief investigator (CI), trial coordinator, funding coapplicants and two members of patient group will bring independent representation. The TSC will take on some aspects of pharmacovigilance. There will be no data monitoring committee as this is a small feasibility study with a short duration and short follow-up, and pharmacovigilance is a component of the study. The Trial Management Group (TMG) will consist of the CI, funding coapplicants, representative of the clinical trials units including trial coordinator, data manager and statistician. They should meet every 2 months to ensure all practical details of the trial are progressing well and adequate targets are met. The patient group consists of patients (and family) from Southend University hospital with chronic kidney disease and suffer from acute gout attacks.

## Interim analysis
There will be no interim analysis as this is a feasibility study, the study procedures will be regularly reviewed at the trial steering committee.

## Harms
Reporting of adverse events will start at the point of consent and reporting for adverse reaction starts at first IMP dose until 7 days' post-treatment. At each contact with the subject during the treatment period, the investigator must seek information on adverse events by specific questioning and, as appropriate, by examination. All clearly related signs, symptoms and abnormal diagnostic procedures should be recorded using the National Cancer Institute (NCI) common terminology criteria for adverse events (CTCAE) V.4.0 event terms and grading. The clinical course of each event should be followed until resolution or stabilisation.

Reporting will follow local operating procedure and study protocol with oversight from the coordinating centre. Non-serious adverse events or reactions will be recorded in the study file with follow-up; and reported for review by the coordinating investigator within a month. All SAEs*/ suspected unexpected serious adverse reaction (SUSARs) must be recorded on the SAE form/ adverse event clinical research form (AECRF) form and faxed to the sponsor within 24 hours of the research staff becoming aware of the event. Any change of condition or other follow-up information should be faxed to the sponsor as soon as it is available or at least within 24 hours of the information becoming available. Events will be followed up until the event has resolved or a final outcome has been reached. All SAEs assigned by the PI or coordinating investigator as both suspected to be related to IMP treatment and unexpected will be classified as SUSARs

and will be subject to expedited reporting to the Research Ethics Committee and Medicines and Healthcare Products Regulatory Agency (MHRA). The CI will complete the Council for International Organisations of Medical Sciences form and if warranted, an investigator alert may be issued, to inform all investigators involved in any study with the same drug (or therapy) that this serious adverse event has been reported.

Potential events/reactions reported with anakinra consist of neutropaenia, infection and less serious ones consist of injection site reactions such as erythema, pruritus and rash. Potential events/reactions with steroid treatment consist of worsening hypertension, blood sugar reading/uncontrolled diabetes and clinical signs of fluid retention. Pre-existing conditions should not be reported as AE unless the condition worsens by at least CTCAE grade during trial. The condition must be reported in the pretreatment section of the eCRF, if symptomatic at the time of entry or under concurrent medical conditions if asymptomatic. Given the potential of clinical events related to underlying comorbid disease burden of potential participants, events that are recognised and expected complications of the condition are exempt from normal reporting procedure, unless they are of an unexpected severity. Common expected events in these patients may consist of worsening complications of chronic kidney disease, heart failure, hypertension, diabetes complications, underlying cardiac disease with complications such as ACS, arrhythmia, hypotension, tachycardia and worsening oedema. Other exceptions to (serious) adverse events SAEs or serious adverse reactions (SARs) reporting consist of routine treatment or monitoring of the studied indication not associated with any deterioration in condition, associated with any deterioration in condition, for example, preplanned hip replacement operation which does not lead to further complications, any admission to hospital or other institution for general care where there was no deterioration in condition, treatment on an emergency, outpatient basis for an event not fulfilling any of the definitions of serious as given above and not resulting in hospital admission, any death or hospitalisation due to fall or fracture, any death or hospitalisation due to exacerbation of an existing medical comorbid condition.

## Auditing

The study will be subject to monitoring, inspection and audit by Anglia Ruskin Clinical Trials Unit, Southend University Hospital NHS Foundation Trust acting as the Sponsor and other regulatory bodies to ensure adherence to Good Clinical Practice (GCP).

## ETHICS AND DISSEMINATION

The study will be conducted in accordance with the principles of GCP. A favourable ethical opinion was obtained from the London-Central Research Ethics Committee, reference number 15/LO/1922. Results will be published in peer-reviewed journal and disseminated at international conferences.

## Protocol amendments

Protocol amendments to the protocol or study documents will be submitted to the REC and can only be implemented once approval has been obtained. Amendments to the clinical trials authority (CTA) or documents that supported the original CTA application will be notified to the MHRA. Amendments will be tracked in the protocol appendix and the version of the protocol will be updated.

## Consent

A clinician in the research team will obtain informed consent from potentially eligible participants. Patients who are unable to consent for themselves for suitability for the trial will not be approached as obtaining patient reported outcome measures will not be practical. If verbal translation is needed, this should be via a hospital interpreter or a personal interpreter. Telephone interpretation services are not acceptable and written material will not be provided in various languages for this feasibility study. This study utilises questionnaire surveys that have not been validated in different languages. There is also a qualitative element to the study where lack of required language skill may prove to be difficult. Time is limited for potential participants to be considering participation and any delays in seeking consent and translation services that may prolong treatment of the patient in acute pain have also to be considered.

Written material consisting of participant information leaflet and consent documentation (see online supplementary material) that has been approved by the Research Ethics Committee and will be compliant with GCP, local regulatory and legal requirement. This is a feasibility study and the costs to undertake centrally commissioned translated documents may be too high. This may be something to consider for the subsequent definitive study.

There will be opportunity for the participant to ask questions to the PI or a member of the research team. Potential participants can have up to 24 hours to consider the information and their participation. However, treatment cannot be withheld during this period and exclusion criteria will be applied if duration of attack is longer than accepted or if contraindicatory medication is administered.

## Ancillary studies

There will be a tertiary/exploratory study where patient's serum and urine will be stored for future analysis at the end of the study. Testing will be an extension of routine laboratory testing to look for markers of inflammation in an exploratory setting.

## Confidentiality

All investigators and trial site staff must comply with the requirements of the Data Protection Act 1998 with regards to the collection, storage, processing and disclosure of personal information and will uphold the Act's

core principles. Information with regards to study patients will be kept confidential and managed in accordance with the Data Protection Act, NHS Caldicott Guardian, The Research Governance Framework for Health and Social Care and Research Ethics Committee Approval.

Data will be stored for 5 years before being destroyed. The chief investigator in the data custodian.

## Declaration of interest

The coordinating investigator has not obtained any personal grants, a investigator sponsored study grant of £10 000 and provision of Anakinra and its equivalent placebo is being provided for the study by the manufacturer's. This fund will be used to cover any predominantly pharmacy and logistics costs.

## Access to data

Only the data manager will have access to data and coordinate access to relevant researchers. The health economist will have access to health economic data set and qualitative researcher will have access of patient demographics to obtain a broad sample.

## Ancillary and post-trial care

Acute gout attack episodes will be treated as part of this trial. If participants get another acute attack of gout, they will not be eligible to entry into the study. The participating team could consider using information with regards to treatment to avoid and available treatment options, but no information will be made available prior to the end of the trial.

## Dissemination policy: trial results and authorship

The trial report will be used for publications and presentation at scientific meetings. All publications and presentations will be reviewed by the TMG. Authorship will be determined per internationally agreed criteria for authorship. Funding bodies will be declared in publications. The full report will be made available directly from the chief investigator or from published material. Supplemental material linked with publication will consist of the trial protocol.

**Author affiliations**
[1]Department of Renal Medicine, Southend University Hospital NHS Foundation Trust, Prittlewell Chase, Southend, Essex, UK
[2]Clinical Trials Unit, Anglia Ruskin University, Bishops Hall Lane, Chelmsford, UK
[3]Norwich Medical School, University of East Anglia, Norwich, UK
[4]School of Health and Human Sciences, University of Essex, Wivenhoe Park, Colchester, UK
[5]The Princess Alexandria Hospital NHS Trust, Harlow, Essex, UK
[6]West Suffolk Hospital NHS Foundation Trust, Bury Saint Edmunds, Suffolk
[7]Lister Hospital, East and North Herfordshire NHS Trust, Corey Mills Lane, Stevenage, Hertfordshire, UK
[8]Centre for Experimental Medicine and Rheumatology, William Harvey Research Institute, Barts and the London, London, UK
[9]Primary Care and Health Sciences, Keele University, Keele, Staffordshire, UK

**Contributors** GB conceived the idea, designed the study, wrote the protocol and the manuscript. BD helped develop the design and the protocol. MA and ER helped refine the design, protocol and the manuscript. JS designed the qualitative aspect of the study. MP undertook the statistical aspect of study design. TM designed the economic evaluation aspect of the study. MH and TP helped developed the logistical aspects of trial involving the trial unit. CM, PH, KA, TM, SB, CP helped refine the study. All authors approved the final manuscript.

**Funding** This paper presents independent research funded by the NIHR under its Research for Patient Benefit (RfPB) Programme (Grant Reference Number PB-PG-0614-34090). The views expressed are those of the author(s) and not necessarily those of the NHS, the NIHR or the Department of Health. There is an additional investigator initiated study fund of £10 000 from SOBI to help with pharmacy logistics.

**Competing interests** None declared.

**Provenance and peer review** Not commissioned; externally peer reviewed.

**Data sharing statement** This manuscript is a study protocol and no data as yet is available.

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
