## [Reviewer comments · BMJ Open]

ARTICLE DETAILS

TITLE (PROVISIONAL)	Study protocol for a feasibility randomised multi-centre, double-blind, double-dummy controlled trial of Anakinra, an interleukin-1 receptor antagonist versus intramuscular methylprednisolone for acute gout attacks in patients with chronic kidney disease: ASGARD
AUTHORS	Balasubramaniam, Gowrie; Parker, Trisha; Turner, David; Parker, Mike; Scales, Jonathan; Harnett, Patrick; Harrison, Michael; Ahmed, Khalid; Bhagat, Sweta; Marianayagam, Thirupathy; Pitzalis, Costantino; Mallen, Christian; Roddy, Edward; Almond, Mike; Dasgupta, Bhaskar

VERSION 1 - REVIEW

REVIEWER	Puja Khanna University of Michigan, Ann Arbor, Michigan USA
REVIEW RETURNED	27-Apr-2017

GENERAL COMMENTS	This is an excellent and well written paper addressing rather important design elements of a clinical trial which has been long time coming. Anakinra is already used extensively in the inpatient setting for acute gout and also in renally impaired patient however data from a RCT is obviously lacking and this protocol will shed light on the safety and feasibility of the drug. Anectodally I have used the drug in >50 patients without any issues or AEs.
--

REVIEWER	Alexander So CHUV Switzerland Consultant for SOBI pharma
REVIEW RETURNED	15-May-2017

GENERAL COMMENTS	You made no mention of preparation of placebo Anakinra - will this be supplied by the manufacturer of Anankinra or will that be prepared separately by the pharmacy of your institution ?
---

VERSION 1 – AUTHOR RESPONSE

Our thanks to reviewer 1 for their kind words and comments.

With regards to reviewer 2, in page section under the section Implementation, we state that "Anakinra and its placebo will be ready made by the manufacturer." This has been changed to specifically state "Anakinra and its placebo will be supplied by the manufacturer."